# Volatile Profiling of *Pleurotus eryngii* and *Pleurotus ostreatus* Mushrooms Cultivated on Agricultural and Agro-Industrial By-Products

**DOI:** 10.3390/foods10061287

**Published:** 2021-06-04

**Authors:** Dimitra Tagkouli, Georgios Bekiaris, Stella Pantazi, Maria Eleni Anastasopoulou, Georgios Koutrotsios, Athanasios Mallouchos, Georgios I. Zervakis, Nick Kalogeropoulos

**Affiliations:** 1Department of Dietetics-Nutrition, School of Health Science and Education, Harokopio University of Athens, El. Venizelou 70, Kallithea, 176 76 Athens, Greece; dtagkoul@hua.gr (D.T.); stellapant15@gmail.com (S.P.); maria_anastaso@outlook.com (M.E.A.); 2Laboratory of General and Agricultural Microbiology, Agricultural University of Athens, Iera Odos 75, 11855 Athens, Greece; giorgosbekiaris@yahoo.gr (G.B.); georgioskoutrotsios@gmail.com (G.K.); zervakis@aua.gr (G.I.Z.); 3Department of Food Science and Human Nutrition, Agricultural University of Athens, Iera Odos 75, 11855 Athens, Greece; amallouchos@aua.gr

**Keywords:** mushrooms, *Pleurotus ostreatus*, *Pleurotus eryngii*, volatiles, headspace solid phase microextraction, cooking, pyrazines, principal component analysis

## Abstract

The influence of genetic (species, strain) and environmental (substrate) factors on the volatile profiles of eight strains of *Pleurotus eryngii* and *P. ostreatus* mushrooms cultivated on wheat straw or substrates enriched with winery or olive oil by products was investigated by headspace solid-phase microextraction coupled with gas chromatography-mass spectrometry (HS-SPME-GC-MS). Selected samples were additionally roasted. More than 50 compounds were determined in fresh mushroom samples, with *P. ostreatus* presenting higher concentrations but a lower number of volatile compounds compared to *P. eryngii.* Roasting resulted in partial elimination of volatiles and the formation of pyrazines, Strecker aldehydes and sulfur compounds. Principal component analysis on the data obtained succeeded to discriminate among raw and cooked mushrooms as well as among *Pleurotus* species and strains, but not among different cultivation substrates. Ketones, alcohols and toluene were mainly responsible for discriminating among *P. ostreatus* strains while aldehydes and fatty acid methyl esters contributed more at separating *P. eryngii* strains.

## 1. Introduction

Mushrooms have been considered a delicacy for centuries, due to their distinct texture, aroma and exceptional flavor, while pertinent research evidenced unique nutritional and health-promoting benefits [1,2,3]. Mushrooms’ popularity steadily increased during the last decades, leading to a 30-fold increment of the world production of edible mushrooms, with *Lentinus edodes* and *Pleurotus* (“oyster”) spp. accounting for over 40% of the respective total supply [4]. *Pleurotus* (oyster) mushrooms are low energy, low fat, low sodium and cholesterol-free food items, being at the same time rich in proteins, minerals, functional polysaccharides like chitin and β-glucans, and water-soluble vitamins. In addition, they contain health-promoting bioactive microconstituents like ergosterol (provitamin D2), phenolic acids, the antioxidant amino acid ergothionein and lovastatin [5,6]. Currently, several species of *Pleurotus* are produced worldwide (e.g., *P. ostreatus*, *P. pulmonarius*, *P. eryngii*, *P. djamor* and *P. citrinopileatus*), by exploiting a wide range of lignocellulosic residues as substrates [7]. Of particular interest is their cultivation on olive mills and wineries by-products (i.e., materials that are difficult to manage due to their high content in organic compounds, notably lipids and polyphenolics), which is in line with the concepts of waste biodegradation and valorization [5,8].

Volatile compounds emitted by fungi are not only associated with key-functions related to growth physiology and mycelium interactions [9,10], but they also add to the quality of edible mushrooms [11,12]. They are additionally formed as a result of post-harvest processing, especially after thermal treatment [13,14]. The characteristic odor of mushrooms is the combined result of several groups of compounds, such as alcohols, aldehydes, ketones, acids, hydrocarbons, esters, and heterocyclic, aromatic and sulfur compounds [12,15,16,17]; their relative abundance may vary among species, strains, stage of growth and part of mushroom examined (i.e., cap, gills or stipe) [17,18,19,20].

More than 110 volatile compounds have been identified so far in mushrooms; among them, eight carbon (C8) compounds account for 44–97% of volatiles fraction, with 1-octen-3-ol, 3-octanol, 3-octanone, 1-octanol and 1-octen-3-one being detected in most cases [2,15,21,22]. In addition, of those identified only 3% are odor-active and can contribute to the aroma or flavor of foods [12,23].

Common laboratory methods for the isolation of aroma compounds include supercritical extraction, simultaneous distillation-extraction, dynamic headspace, and headspace solid phase microextraction (HS-SPME); the latter is an easy, rapid, relatively simple technique that allows the isolation and concentration of volatile analytes in a solvent free, environmentally friendly, manner [2,21,22,24]. Coupled with gas chromatography–mass spectrometry (GC-MS), HS-SPME is considered a powerful tool with high reproducibility and sensitivity [21,24,25,26,27].

In the present study, we investigated the influence of genetic (species, strain) and environmental (substrate) factors on the profile of volatile compounds produced by eight *P. ostreatus* and *P. eryngii* strains (four per species), which were cultivated on substrates composed of various agricultural and agro industrial by-products. The main objective was to verify whether a discrimination based on volatile aroma compounds was possible among species, strains and/or substrates. Volatiles were extracted from fresh fruit bodies with HS-SPME and analyzed by GC-MS. The same analytical scheme was applied to selected samples to assess the influence of cooking (roasting) on volatiles’ profiles.

## 2. Materials and Methods

### 2.1. Standards

Internal standard 4-methyl-1-pentanol was purchased from Sigma-Aldrich (Merck, Darmstadt, Germany), C8-C40 alkanes calibration standard for calculation of linear retention indices (RI) and a standard mixture of 37 fatty acid methyl esters (FAME) were obtained from Supelco (Supelco Inc., Bellefonte, PA, USA).

### 2.2. Biological Material and Mushroom Cultivation Substrates

The following *Pleurotus eryngii* and *P. ostreatus* strains were used for the purposes of this study: CS1, CS2, CS3 and LGAM106 for the former, and CS4, CS5, CS6 and LGAM3002 for the latter species. All strains are maintained in the Culture Collection of the Agricultural University of Athens (Laboratory of General and Agricultural Microbiology, Athens, Greece). Mushrooms were cultivated on three substrates consisting of (i) wheat straw (WS, as control), (ii) wheat straw in 1:1 *w*/*w* ratio with grape marc (GM), and (iii) olive leaves in 3:1 *w*/*w* ratio with two-phase olive mill wastes (OL). WS were obtained from the Thessaly region (central Greece), GM from a winery in the Nemea area (northeast Peloponnese, Greece) and OL from an olive-oil mill in Kalamata (southwest Peloponnese, Greece). Spawn and substrate preparation methodologies as well as mushroom cultivation conditions were as previously described [5,28].

### 2.3. Extraction of Aroma Volatile Compounds

Aroma volatile compounds were isolated by head space solid phase microextraction (HS-SPME) and identified and quantitated by GC-MS. For this purpose, immediately after harvest, approximately 50 g of fresh mushroom samples from each treatment (in triplicates) were wrapped in aluminum foil, sealed in plastic bags and kept at −40 °C until analysis, which was conducted within one week. Prior to analysis, samples were thawed, finely chopped and approximately 1 g was weighed into 15 mL screw capped glass vials (Supelco, Bellefonte, PA, USA), followed by the addition of 4 mL of saturated aqueous NaCl solution to inhibit enzymatic degradation and boost the release of volatiles [17,29] and 50 μL of internal standard (4-methyl-1-pentanol) methanolic solution (μg/mL). A magnetic stir bar was immersed in each vial, which was sealed with PTFE-faced/silicone septum (Supelco, Bellefonte, PA, USA). Sealed SPME vials were kept in a water bath at 40 °C, under magnetic stirring (250 rpm, Phoenix Instrument RSM-10 HS/HP, Garbsen, Germany) for 5 min to equilibrate. Then the SPME assembly was inserted, and fiber was exposed in the headspace and left to incubate for 40 min under continuous stirring. The volatiles were extracted with 50/30 μm divinylbenzene/carboxen/polydimethylsiloxane (DVB/CAR/PDMS) StableFlex fibers (Supelco Inc., Bellefonte, PA, USA), reported exhibiting good selectivity and high efficiency for the extraction of volatiles from wild and cultivated mushrooms [15,17,27,30,31,32]. They are additionally more selective for the identification of aldehydes such as methional and phenylacetaldehyde, considered important for mushroom flavor [32].

### 2.4. Gas Chromatography-Mass Spectrometry Analysis

An Agilent GC 6890N gas chromatograph (Waldron, Germany) coupled with HP5973 Mass Selective detector (electron impact 70 eV) and split—splitless injector, with a specific 0.75 mm i.d. liner provided by Supelco (Bellefonte, PA, USA) was used for volatiles profiling. Following incubation, the SPME assembly was removed from the vial and inserted into the GC injection port, where it remained for 20 min at 220 °C to desorb volatiles. The separation was achieved in a J&W 122-7032 DB-WAX column (30 m, 0.25 mm i.d., 0.25 μm film thickness). High purity helium was the carrier gas at a constant flow of 1.0 mL/min. GC operated in 1:1 split mode and mass detector operated at full scan mode covering 33–350 *m*/*z* mass range. The injector and transfer line temperatures were kept at 220 °C. The oven temperature program was: initial temperature 35 °C for 2 min, followed by a ramp of 20 °C /min to 100 °C, and finally 5 °C /min to 240 °C. The MS data were obtained by the mass detector operating at full scan in the range of *m*/*z* 35–350.

### 2.5. Identification and Semi-Quantification of Volatile Compounds

The identification of chromatographic peaks was performed by comparing the mass spectrum of each compound with the Wiley 275 (Wiley, New York, NY, USA) and NIST 98 (NIST MS search v6.1d) mass spectral databases. Further confirmation was performed by calculating the Kovats linear retention indices (RIs), using n-alkanes (C8–C40) standard (Supelco, Bellefonte, PA, USA) as external reference [33] and comparing the values obtained with those reported in the literature [34]. Among the compounds detected, only those exhibiting mass spectra matching qualities higher than 90% and their calculated RI did not differ by more than ±15 from the values available in public domain databases were included in the respective Table 1, Appendix A. For the aroma description of volatiles, “The Good Scents Company Information System” database was used [35]. Semi-quantification was performed based on the peak area and amount (μg) of the internal standard.

### 2.6. Cooking of Mushroom Samples

Cooking (roasting) of mushrooms was carried out in a domestic oven. For this purpose, approximately 100 g of fresh *P. ostreatus* LGAM3002 and *P. eryngii* CS3, cultivated on WS, were weighed, placed in clean Pyrex dishes and roasted at 180 °C for 10 min without aeration. The Pyrex dish was subsequently removed from the oven, covered with aluminum foil and allowed to cool for 5 min. Mushrooms were weighed before and after cooking to estimate water loss. Cooking procedure was conducted in triplicate. The pre-treatment and analysis of roasted samples was the same as that applied for the fresh ones.

### 2.7. Total Lipids Content

Mushrooms lipid content was measured in freeze-dried, pulverized samples by the colorimetric sulfo-phospho-vanillin reaction, employing commercial sunflower oil as lipid standard [36,37].

### 2.8. Statistical Analysis

Analyses were performed in triplicate and data are presented as mean ± standard deviation. Differences between means were established by one way analysis of variance (ANOVA) and Duncan’s *t*-test (*p* < 0.05) with the SPSS software (SPSS for Windows, version 21.0, SPSS Inc., Chicago, IL, USA). Principal component analysis (PCA) was performed on the entire volatile compounds profile set, as well as on the volatile compounds’ classes, to attain an overview of possible associations between volatiles and species/strains, raw and roasted mushroom samples and/or different cultivation substrates. Mean centering was only applied on the respective data sets. R-studio 1.0.136/R3.3.3 loaded with the “ade4” [38] and “adegraphics” [39] packages, were used for PCA.

## 3. Results and Discussion

### 3.1. Total Volatiles Contents

Overall, according to the criteria set at paragraph 2.5, more than 80 compounds were identified in mushrooms of the present study, with 24 of them detected only in roasted samples (Table 1). Fresh *P. ostreatus* mushrooms contained a lower number of volatile compounds compared to fresh *P. eryngii* ones (36 vs. 56 compounds). In accordance with our observation, Yin et al. (2019) [40] reported that *P. ostreatus* had the lower number of volatile compounds among six *Pleurotus* species. However, one should not overlook that the concentrations of volatile compounds alone are not a sufficient base for studying the aroma profile of mushrooms, as different volatile compounds may present significantly different odor thresholds [30].

Nevertheless, concentrations of the volatiles (i.e., compounds included in Table 2 and Table 3) were higher in *P. ostreatus* compared to *P. eryngii* (1684 ± 591 vs. 965.4 ± 321.5 μg/g f.w), in agreement with the data reported by Jung et al. (2019) [2]. The compounds detected in fresh samples are classified as aldehydes, alcohols, ketones, alkanes, fatty acid methyl esters (FAME) and terpenes. Moreover, the heterocyclic 2-pentyl-furan and the aromatic hydrocarbons toluene and ethylbenzene were also detected. Regarding the strains studied, *P. eryngii* LGAM106 and *P. ostreatus* CS5, CS6 and LGAM3002 presented the higher total volatiles content. In most cases, cultivation on GM and/or OL resulted in enhanced volatiles production however this pattern was not consistent (Figure 1).

### 3.2. Aldehydes

Fifteen aliphatic and aromatic aldehydes were identified in fresh *Pleurotus* samples, 12 of them in *P. ostreatus* and 14 in *P. eryngii* (Appendix A). Aliphatic aldehydes are biochemically derived from fatty acids, whereas the aromatic ones derive from amino acids [41]. Aldehydes confer a fresh, floral grassy and fatty aroma [17,29], and are characterised by low odor thresholds, i.e., they can be perceived even in small concentrations [40]. As a general trend, three out of four *P. eryngii* strains contained more volatile aldehydes than *P. ostreatus* strains, while this was also the case for mushrooms cultivated on WS and/or GM (Figure 1). Hexanal was the predominant aldehyde in both species (in accordance to the findings of a recent study [2]), followed by 2-octenal, n-octanal and 2-heptenal. *P. eryngii* contained three times more benzaldehyde than *P. ostreatus* (17.72 ± 11.17 vs. 5.58 ± 2.96 μg/g f.w.). According to Mau et al. (1998) [41], benzaldehyde was the major volatile compound in fruitbodies of *P. eryngii*. This aromatic aldehyde derives from the catabolism or oxidative degradation of phenylalanine [29,42,43] which is present in the *Pleurotus* mushrooms studied [44]. Benzaldehyde together with phenylacetaldehyde and short-chain aldehydes (pentanal, hexanal, octanal, nonanal, *trans*-2-heptenal) detected in the mushrooms studied (Table 1), are oxidative degradation products of oleic and linoleic acids; according to a recent report, they are among the major contributors of edible oil flavors during ambient storage [45]. Heptanal, 2-hexenal, phenylacetaldehyde, 2,4-nonadienal, and 2,4 decadienals were also present in the volatiles of both mushroom species, at slightly higher concentrations in *P. eryngii*; finally, nonanal, undecenal and 3-dodecenal were detected only in *P. eryngii,* while 2-phenyl-2-butenal was recorded only in *P. ostreatus*. The aldehydes identified in this study have been previously reported in several mushroom species [2,17,29,40].

### 3.3. Alcohols

Seven aliphatic and one aromatic alcohols were identified in fresh *Pleurotus* samples, (Appendix A), their overall concentration being almost double in *P. ostreatus* compared to *P. eryngii* (1200 ± 458 vs. 666.9 ± 252.3 μg/g f.w.). In all strains studied, 1-octen-3-ol, predominated, i.e., 650.7 ± 212.4 and 622.0 ± 225.2 μg/g f.w. in *P. ostreatus* and *P. eryngii*, respectively. 1-Octen-3-ol, the so called “mushroom alcohol” [46], is a key odor compound deriving from linoleic acid through oxygenation and subsequent cleavage of the respective hydroperoxide [47]. It is responsible for the earthy, green, oily, fungal aroma of mushrooms, identified as a major aroma compound in many mushroom species [2,17,40,46]. In addition, 3-octanol, 2-octen-1-ol, 1-octanol, 2-ethyl-1-hexanol, and hexanol were also detected in both species, while phenylethyl alcohol was found only in *P. eryngii*. Alcohols provide aromas, similar to those of aldehydes. Unsaturated alcohols have a lower odor threshold than the saturated ones, therefore they affect to a greater extent the overall aroma [40].

### 3.4. Ketones

Ketones are products of fatty acids metabolism [42,48]. In the present study seven aliphatic and one aromatic ketones were recorded in *P. eryngii* mushrooms but only four (3-octanone, 1-octen-3-one, 2,3-octanedione and 2-undecanone) in *P. ostreatus* strains; however, the latter contained a higher amount of ketones (266.9 ± 101.9 μg/g f.w. in *P. ostreatus* vs. 68.59 ± 20.60 μg/g f.w. in *P. eryngii*) (Table 2). Nonetheless, no clear and consistent distribution pattern is observed when the cultivation substrates are taken into account (Figure 1). Among the ketones detected, 3-octanone is predominant in both species at concentrations of 51.01 ± 15.44 and 249.9 ± 98.8 μg/g f.w. in *P. eryngii* and *P. ostreatus*, followed by 1-octen-3-one, 2-undecanone and 2,3-octanedione in lower concentrations, while acetophenone, 2-octanone, 6-methyl-5-hepten-2-one, and 3-octen-2-one were detected only in *P. eryngii* strains (Appendix A). Regarding their aromas, 3-octanone is described having a sweet, fruity, musty and lavender aroma [21], while 1-octen-3-one represents the dried-mushroom flavor imparting wet ground smell [40,49]. They both possess low odor detection thresholds, and are considered important contributors to the aroma of mushrooms [21,40]. The ketones detected in the present study have been reported among volatiles of various edible mushroom species [2,18,20,29,40,50,51].

### 3.5. Fatty Acid Methyl Esters (FAME)

FAME originate from fatty acids metabolism [42,48] and impart fatty and fruity aromas, with very low odor thresholds [17]. In the present study, seven FAME were identified in *P. eryngii*, and six in *P. ostreatus*, their total concentrations being 65.79 ± 40.31 vs. 13.91 ± 8.49 μg/g f.w., in line with their respective lipids content (7.29 ± 0.4 vs. 3.73 ± 0.2 mg/g f.w.). Methyl palmitate was the predominant FAME in both species (31.44 ± 20.62 vs. 6.02 ± 3.11 μg/g f.w. in *P. eryngii* and *P. ostreatus*) followed by methyl linoleate and methyl oleate (Appendix A). Methyl pentadecanoate, methyl laurate and methyl myristate were also detected at lower concentrations, while methyl stearate was found only in *P. eryngii* samples. There is a clear difference regarding the levels of FAME among the species studied since volatiles of *P. eryngii* strains contain more FAME than *P. ostreatus* (Figure 1). Regarding the influence of cultivation substrates, volatiles of *P. eryngii* mushrooms from OL and GM contained more FAME, whereas no clear pattern was evident in *P. ostreatus* (Figure 1). The FAME identified in this study have been reported in various *Pleurotus* spp. [51,52], *Morchella importuna* [17], *Agaricus bisporus* [27], and eleven wild edible mushrooms [31].

### 3.6. Alkanes

Alkanes confer woody, terpene and waxy aromas to mushrooms [17]. Thirteen alkanes were detected in volatiles of *P. eryngii* and five in *P. ostreatus* mushrooms, the most abundant being undecane (15.15 ± 4.26 vs. 18.03 ± 8.01 μg/g f.w.) (Appendix A). No specific trend regarding their distribution among species, strains or cultivation substrates was observed. The alkanes detected in the present study have been also identified in the volatiles of various mushroom species [17,19,27,40,50,52,53,54].

### 3.7. Terpenes

Three terpenes were identified in *P. eryngii,* but only one in *P. ostreatus* strains. The monoterpene limonene was present in three out of four *P. ostreatus* strains, at concentrations ranging from 0.09 ± 0.02 to 2.59 ± 1.27 μg/g f.w. in *P. ostreatus* and 0.34–0.45 μg/g f.w. in *P. eryngii* CS1 grown on GM and OL (Appendix A). Limonene has been reported in *P. eryngii* and *P. cystidiosus* [52] and other edible mushrooms, like *Tricholoma matsutake* [18,55], *A. bisporus* [19], various species growing in the wild [20], several truffles [53] and *L. edodes* [54]. The isoprenoid hydrocarbons pristane (2,6,10,14-tetramethylpentadecane) and phytane (2,6,10,14-tetramethylhexadecane) were detected in *P. eryngii* CS3 and LGAM106, cultivated on GM and OL at 3.92–15.46 μg/g f.w. (Appendix A). Phytane has been also detected in hot water extract of tiger milk mushroom (*Lignosus rhinocerus*) [56], while no literature data concerning pristane in edible mushrooms were found. Terpenes confer a pleasant camphoric aroma as well as exotic fruit notes (e.g., immature mango fruit) [53].

### 3.8. Other Compounds—Toluene

The aromatic hydrocarbons toluene and ethylbenzene as well as the heterocyclic 2-pentylfuran, not belonging to the groups mentioned above, are discussed separately. Toluene was present in all strains of both species at 17.05 ± 8.11 vs. 104.5 ± 38.8 μg/g f.w. in *P. eryngii* and *P. ostreatus*, respectively. Toluene derives from unsaturated fatty acids and has been reported in wild edible species [15], freeze-dried truffle (*Tuber aestivum*) [57], and other food items like cooked beef [58,59] and fresh Mediterranean fish and shellfish [60]. Toluene exhibited a distinct distribution among *Pleurotus* volatiles, its concentrations being 3–13 times lower in *P. eryngii* when compared to *P. ostreatus* (Figure 1, Appendix A). Ethylbenzene was detected at low concentrations only in *P. eryngii*. It has been reported in food items like cooked beef [58] and fermented cucumbers [61] but—to the best of our knowledge—not in edible mushrooms. The heterocyclic compound 2-pentylfuran is formed by the autoxidation of linoleic acid [62]; its odor is described as green, earthy and meaty [17,40]. It was detected at low concentrations in *P.*
*eryngii* and *P. ostreatus* (0.17–0.94 μg/g f.w.), and it did not exhibit any specific distribution pattern. It was identified in *Pleurotus* [40] and other edible mushrooms [2,15,17,50].

### 3.9. Eight-Carbon Compounds

Eight-carbon volatile compounds are major contributors to the characteristic flavor of many mushrooms [2,17,21,40,49]. They are oxidation products of linoleic acid via a 10-hydroperoxide intermediate [21,23,40] which represents 60.6–80.6% of fatty acids in *P. eryngii* and *P. ostreatus* [3,7]. They are classified as oxylipins and are involved in a wide range of biological processes [21]. Eight-carbon compounds predominated among the volatiles of all *Pleurotus* strains studied, comprising 78–83% and 84–91% of total volatiles in *P. eryngii* and *P. ostreatus*, following similar distributions with the total volatiles presented in Figure 1. Their concentrations were 776.5 ± 282.0 μg/g f.w. in *P. eryngii* and almost doubled in *P. ostreatus* (1498 ± 551 μg/g f.w.) (Table 2). In addition, 1-octen-3-ol predominated in all *P. eryngii* and in three out of four *P. ostreatus* strains, comprising 88–95% and 47–53% of eight-carbon compounds, respectively. Differences were observed among the *Pleurotus* strains studied: (a) three of the eight-carbon compounds detected in *P. eryngii* were absent from *P. ostreatus*, and (b) in three out of four *P. eryngii* strains, the second more abundant eight-carbon compound was 3-octanone followed by 2-octenal, while in three out of four *P. ostreatus* strains it was 3-octanol, followed by 3-octanone (Table 2).

### 3.10. Effect of Cooking on Volatiles’ Profiles

The volatile compounds determined in fresh and roasted mushrooms of *P. eryngii* strain CS3 and *P. ostreatus* strain LGAM 3002 cultivated on wheat straw are shown in Table 3. Besides water loss, roasting caused significant alterations in the volatiles’ profiles for both species. Overall, the number of volatile compounds decreased from 35 to 27 in roasted *P. eryngii* and increased from 35 to 41 in roasted *P. ostreatus* (Table 3). The percent abundance of volatile alcohols decreased by 65% and 77% in roasted *P. eryngii* and *P. ostreatus,* respectively; the same trend was followed by fatty acid methyl esters, which were almost eliminated (they decreased by 95% and 97% in roasted *P. eryngii* and *P. ostreatus,* respectively). Similarly, ketones, aldehydes and eight-carbon compounds decreased by 66%, 64% and 65% in *P. eryngii* and by 85%, 9.0% and 81% in *P. ostreatus*. Elimination of FAME during the freeze-drying of *A. bisporus* has been reported by Pei et al. (2016) [27].

In addition, roasting resulted in the formation of nitrogen-containing heterocyclic compounds (17 pyrazines, one pyrrole and one oxazole) and two sulfur compounds, i.e., substances with low odor threshold values, known to enrich food flavor with meat- or roast-like aromas [63]. The predominant compounds in roasted *P. eryngii* were 1-octen-3-ol (mushroom-like flavor), followed by hexanal (green and woody), toluene (sweet, solventy, woody, roasted coffee) and undecane (gasoline-like). In roasted *P. ostreatus* 1-octen-3-ol also predominated, followed by 3-octanol (mushroom, buttery), toluene, 3-octanone (mushroom, ketonic, cheesy) and 3-ethyl-2,5-dimethyl pyrazine (hazelnut, nutty). Several aroma compounds detected in fresh and cooked mushrooms originate from the enzymatic and oxidative decomposition of unsaturated fatty acids or from reactions between amino acids and carbonyl compounds, i.e., the Maillard reactions and the Strecker amino acid degradation which leads to the formation of Strecker aldehydes and other flavor-active compounds [48,64,65]. The extent of these transformations depends on the types of compounds involved, temperature, pH and reaction time [65,66,67]. Regarding the influence of temperature, the volatile compounds produced by lipid oxidation (aldehydes, ketones, and aromatic hydrocarbons like hexanal, octanal, benzaldehyde and toluene) are formed at relatively low temperatures, even at room temperature [45]; this provides an explanation for their presence in the fresh mushroom samples (Table 3). By contrast, Maillard and Strecker reactions take place at higher temperatures. Literature data consider temperatures around 120–130 °C as optimal for the formation of Maillard reaction products with favorable sensory characteristics [68] as well as for the formation of compounds with sulfur and nitrogen groups [69,70], e.g., pyrazines, Strecker aldehydes, dimethyldisulfide and pyrrole (detected in the roasted *Pleurotus* samples; Table 3). It is noteworthy that *Pleurotus* mushrooms contain significant amounts of Maillard reaction precursors, namely free amino acids, reported to comprise 37.1–41.6% of crude protein [44] and glucans, which comprise 40–50% *w*/*w* of the total content on a dry weight basis [71]. In addition, the Strecker aldehydes 2-methylbutanal, 3-methylbutanal, methional, phenylacetaldehyde and benzaldehyde, detected in the roasted samples (Table 3) derive from isoleucine, leucine, methionine and phenylalanine [67,72], i.e., amino acids that are present in the mushrooms studied [44].

Pyrazines are heterocyclic aromatic compounds, formed by the reaction between amines and dicarbonyl compounds through Strecker’s degradation [73,74]. They provide roasted, nutty flavor notes to the aroma of thermally-treated food products, even at very low amounts, because of their very low odor thresholds [75,76]. Strecker aldehydes and/or nitrogen heterocyclic compounds have been repeatedly identified among the volatiles of thermally-dried or heat-extracted samples of *Pleurotus* [42], *Boletus edulis* and *P. ostreatus* [77], *A. bisporus* [78], *Hericium erinaceus* [79], *Boletus edulis* [80], as well as in oils obtained from hydro distillation of *Pleurotus* mushrooms [51,52]. They were also present in cooked *P. ostreatus* [81], cooked wild mushrooms [15] and in *A. bisporus* soup [82]. Interestingly, Strecker aldehydes and/or pyrazines were also detected in freeze-dried *M. importuna* [17], dried *L. edodes* [54] fresh and freeze-dried truffles [29,57]. In the present study, six and 17 pyrazines were detected in roasted *P. eryngii* and *P. ostreatus*, with 3-ethyl-2,5-dimethyl pyrazine predominating at 8.76 and 21.22 μg/g f.w., followed by 2-ethyl-3,5-dimethylpyrazine, 2,5-dimethylpyrazine, 2-ethylmethylpyrazine and 3,5-diethyl-2-methylpyrazine at concentrations ranging between 2.9–6.0 μg/g f.w. (Table 3). Among the pyrazines determined in roasted *Pleurotus* mushrooms, 3-ethyl-2,5-dimethylpyrazine is formed from the thermal treatment of alanine [66], methylpyrazine, ethylpyrazine, 2-ethyl-methylpyrazine, and 2,6-diethylpyrazine originate from the thermal treatment of serine and 2,5-dimethylpyrazine, 2,6-dimethylpyrazine, and 2-ethyl-3,5-dimethylpyrazine from the thermal treatment of threonine [74]. Finally, ethyldimethylpyrazines, also detected in the roasted *Pleurotus* samples, originate from the Strecker degradation of methionine and are considered important flavor contributors in thermally processed foods [65]. Alanine, serine, threonine and methionine are among the amino acids reported in the *Pleurotus* mushrooms studied [44].

It is noteworthy that due to the absence of thermal treatments, pyrazines, pyrrole, oxazole and sulfur compounds were detected only in the cooked samples (Table 3); hence, the volatile profiles of fresh mushrooms obtained in the present study can be considered as representative, in some degree, of the *P. eryngii* and *P. ostreatus* aroma.

The sulfur compounds methional and dimethyl disulfide were detected in the roasted *Pleurotus* samples (Table 3). Methional is the Strecker aldehyde of methionine and is eventually decomposed to form dimethyl disulfide, which contributes further to the overall flavor development [65,67]. Methional is responsible for the aroma of cooked potatoes [67] and is the key aroma-active compound in cooked *T. matsutake* mushrooms [55], while both sulfur compounds were reported in *A. bisporus* soup [82].

### 3.11. Principal Component Analysis (PCA)

Principal component analysis (PCA) was performed on the entire volatile compounds profile of *P. eryngii* and *P. ostreatus* strains, in order to investigate the existence of any groupings or associations (Figure 2). The first two principal components, explaining 98% of the data set variance (i.e., PC1: 75.4% and PC2: 22.6%), allowed a clear discrimination of two *Pleurotus* species and eight strains studied (Figure 2a,b; ellipses drawn at a confidence level of 0.95).

However, due to the high number of identified compounds, it was impossible to acquire some additional information regarding the volatile compounds that were responsible for the species/strain discrimination through the interpretation of their loadings. Therefore, a PCA was performed on the basis of the volatile compounds’ classes, which revealed a clear separation at species and strain level (Figure 3a,b, respectively). Furthermore, the first two principal components explained an even higher percentage (98%) of the data set variance (i.e., PC1: 95.7% and PC2: 2.9%). Separation is taking place in both axes (PC1 and PC2), with *P. eryngii* strains mainly placed in the 4th quartile and *P. ostreatus* strains in the 2nd quartile. The observation of the loadings (i.e., the compound classes) revealed a positive effect of ketones, alcohols and toluene at separating *P. ostreatus* strains (Figure 3c). On the other hand, aldehydes and fatty acid methyl esters presented a stronger effect on the separation of *P. eryngii* strains. Both observations are in agreement with the previously determined content in volatile compounds for these particular species/strains.

PCA was also performed on the basis of volatile compounds’ classes for cooked (roasted) and raw (fresh) *P. eryngii* strain CS3 and *P. ostreatus* strain LGAM 3002 cultivated on wheat straw, to detect any groupings and to acquire additional information. The PCA allowed a clear discrimination of samples both in terms of species/strains as well as of the roasted/raw treatment (Figure 4a,b). As regards the latter, in particular, separation took place across PC1 (X-axis) with the fresh samples being placed on the negative side of PC1 and the roasted on the positive side. By examining the PCA loadings, it appears that pyrazines, sulfur compounds and toluene exert a significant role at separating the roasted samples. However, PCA was not able to discriminate mushrooms according to the cultivation substrate on the basis of the volatile compounds or compound classes studied (data not shown). This possibly indicates a low effect of the cultivation substrates on the mushrooms content in volatile compounds, contrary to the outcomes of our previous works regarding antioxidant properties and bioactive microconstituents like ergosterol, phenolic and terpenic compounds as well as free amino acids in the same mushroom species [5,44].

## 4. Conclusions

The aroma of *Pleurotus* mushrooms is formed by several classes of compounds namely alcohols, aldehydes, ketones, FAME, alkanes and terpenes. Fifty volatile compounds were identified in *P. eryngii* strains, while only 31 in *P. ostreatus*. However, in most cases, *P. ostreatus* presented a higher content in volatile alcohols, ketones and toluene compared to *P. eryngii*, whereas the opposite was established for aldehydes and FAME. Roasting of mushrooms caused partial elimination of existing volatiles and the formation of Maillard reaction and Strecker degradation products, mainly pyrazines, which contribute to the distinctive cooked-mushrooms aroma. Unsaturated alcohols and ketones containing eight carbon atoms predominated among the volatiles of both raw and cooked mushrooms. Principal component analysis performed on the aroma profiles data obtained succeeded in discriminating among *Pleurotus* species/strains and roasted/fresh samples, but not among mushrooms deriving from different cultivation substrates. The findings from the present work are interesting considering that previous studies on the same mushrooms revealed an influence of substrates on the antioxidant activity, β-glucans content and the levels of bioactive microconstituents like ergosterol, phenolic and terpenic acids, and free amino acids [5,45]. The extent to which the mushrooms organoleptic properties are affected by the choice of species/strains or the type of substrate needs to be further investigated by including the determination of additional compounds and/or sensory evaluation.

## Figures and Tables

**Figure 1 foods-10-01287-f001:**
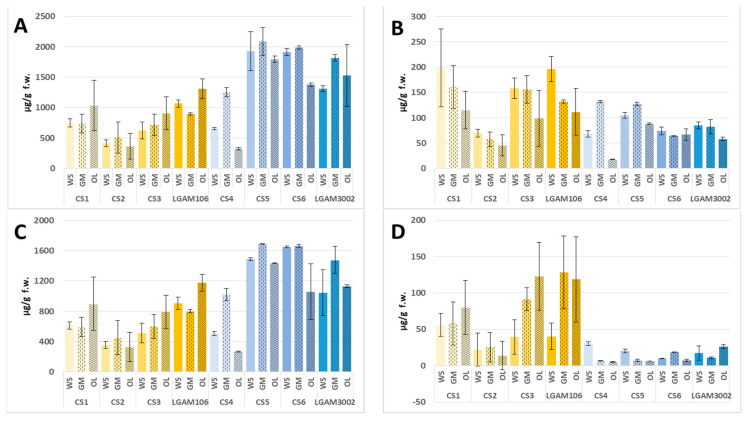
Main volatile classes (μg/g f.w. equivalents of 4-methyl-2-pentanol) in eight strains of *P. eryngii* and *P. ostreatus* grown on three cultivation substrates. (**A**) Eight carbon atoms compounds, (**B**) aldehydes, (**C**) alcohols, (**D**) FAME, (**E**) ketones, (**F**) toluene. Abbreviations f.w.; fresh weight, n.d.: not detected, CS: commercial strains; LGAM: Laboratory of General and Agricultural Microbiology (Agricultural University of Athens, Greece); FAME: fatty acid methyl esters; WS: wheat straw; GM: wheat straw with grape marc (1:1 *w*/*w*); OL: olive leaves with olive mill wastes (3:1 *w*/*w*).

**Figure 2 foods-10-01287-f002:**
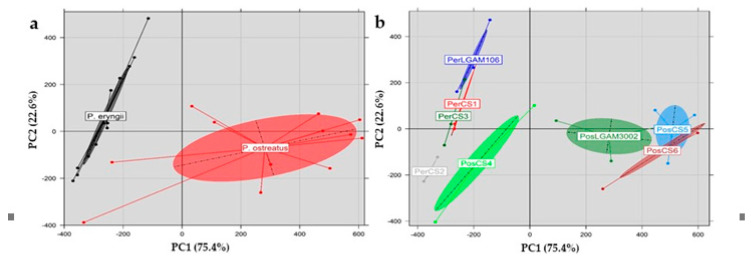
Score plot of principal component analysis and discrimination on the basis of the volatile compounds detected of *Pleurotus* mushrooms at (**a**) species and (**b**) strain level (Pos: *P. ostreatus*; Per: *P. eryngii*).

**Figure 3 foods-10-01287-f003:**
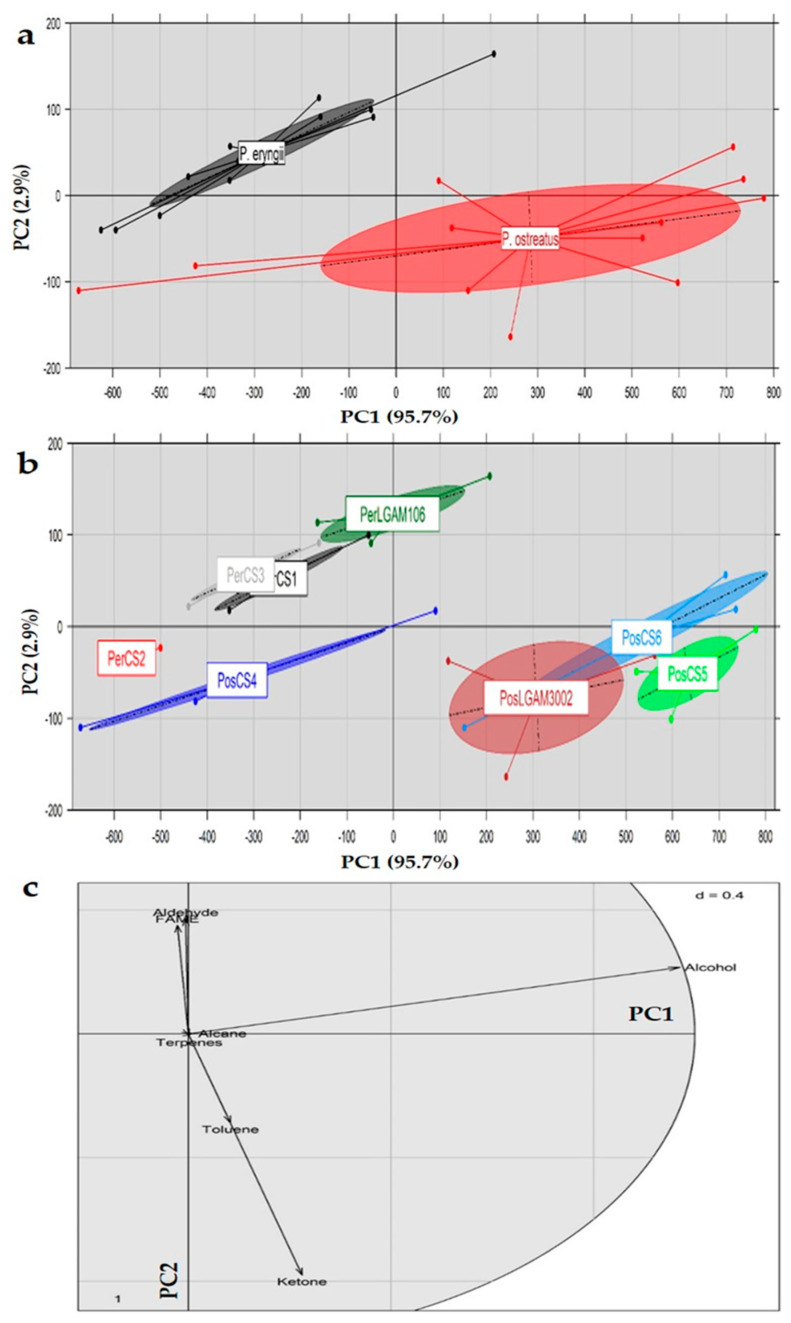
Score plot of principal component analysis and discrimination on the basis of classes of volatile compounds detected in *Pleurotus* mushrooms at (**a**) species and (**b**) strain level, as well as their (**c**) loadings. (Pos: *P. ostreatus*; Per: *P. eryngii*).

**Figure 4 foods-10-01287-f004:**
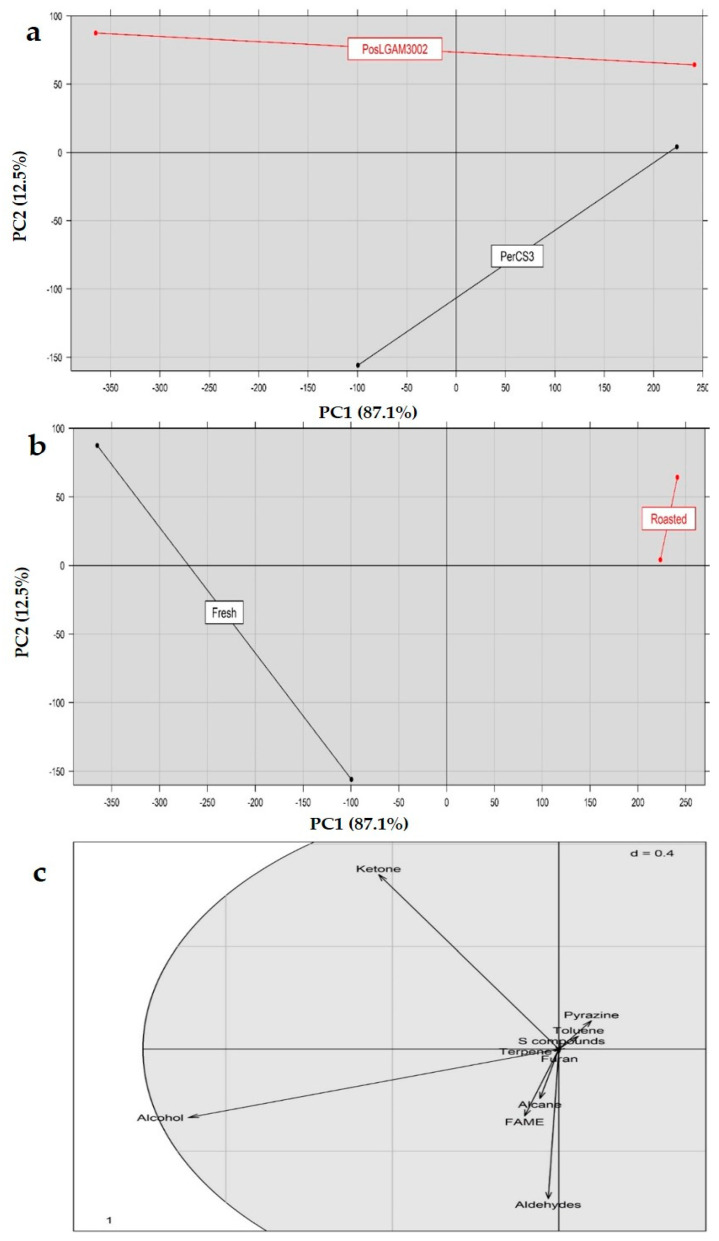
Score plot of principal component analysis and discrimination on the basis of classes of volatile compounds detected in fresh vs. roasted *Pleurotus* mushrooms at (**a**) species/strain level (*P. ostreatus* vs. *P. eryngii*) and (**b**) treatment (fresh vs. roasted) level, as well as their (**c**) loadings (Pos: *P. ostreatus*; Per: *P. eryngii*).

**Table 1 foods-10-01287-t001:** Names, classes, retention times (Rt) and retention indices (RIs) of volatile compounds identified by gas chromatography–mass spectrometry (GC-MS).

No	Class of Compound	Compound Name	Rt (min)	RI	Odor Description
1	Aldehyde	2-methylbutanal *	3.24	908	Musty, chocolate, nutty, furfural and isovaleraldehyde-like with malty and fermented nuances
2	Aldehyde	3-methylbutanal *	3.29	913	Ethereal, aldehydic, chocolate peach, fatty
3	Alkane	1,3-octadiene	3.75	966	
4	Aldehyde	pentanal	3.89	977	Diffusive, fermented, bready, fruity with berry nuances
5	Aromatic compound	toluene	4.58	1048	Sweet
6	sulfur compound	dimethyl disulfide *	4.82	1071	Sulfurous, vegetable, cabbage, onion
7	Aldehyde	hexanal	4.98	1087	Green, fatty, leafy, vegetative, fruity, with a woody nuance
8	Alkane	undecane	5.11	1100	Herbal, woody
9	Aromatic compound	ethylbenzene	5.40	1129	
10	Ketone	2-heptanone	5.89	1178	Cheese, fruity, ketonic, green banana, with a creamy nuance
11	Aldehyde	n heptanal	6.00	1189	Green
12	Heterocyclic compound	2,4,5-trimethyloxazole *	6.07	1196	Nutty, roasted, wasabi, shellfish, mustard, vegetable
13	Alkane	dodecane	6.11	1200	
14	Terpene	limonene	6.16	1205	Terpene, pine, herbal, peppery
15	Aldehyde	trans-2-hexenal	6.37	1224	Sweet, almond, fruity, green, leafy, apple, plum, vegetable
16	Heterocyclic compound	2-pentyl-furan	6.47	1233	Fruity, green, earthy beany with vegetable like nuances
17	Ketone	3-octanone	6.75	1259	Musty, mushroom, ketonic, moldy and cheesy fermented with a green, vegetative nuance
18	Pyrazine	methylpyrazine *	6.83	1267	Nutty, cocoa, roasted, green
19	Ketone	2-octanone	7.09	1291	Musty, ketonic, bleu and parmesan cheese-like with earthy and dairy nuances
20	Aldehyde	octanal	7.14	1295	Aldehydic, waxy, citrus orange with a green peely nuance
21	Alkane	tridecane	7.19	1300	
22	Ketone	1-octen-3-one	7.29	1308	Earthy, metallic, mushroom-like with vegetative nuances of cabbage and broccoli
23	Internal standard	4-methyl-1-pentanol	7.40	1317	
24	Ketone	2,3-octanedione	7.50	1324	Dill, asparagus, cilantro, herbal, aldehydic, earthy, fatty, cortex
25	Pyrazine	2,5-dimethylpyrazine *	7.50	1324	Nutty, peanut, musty, earthy, powdery and slightly roasted with a cocoa powder nuance
26	Aldehyde	2-heptenal	7.60	1332	Green, fatty
27	Pyrazine	2,6-dimethylpyrazine *	7.69	1339	Cocoa, roasted nuts, roast beef, coffee
28	Ketone	6-methyl-5-hepten-2-one	7.72	1341	Fruity, apple, musty, ketonic and creamy with slight cheesy and banana nuances
29	Pyrazine	ethylpyrazine *	7.75	1344	Nutty, musty, fermented, coffee, roasted, cocoa and meaty nuances
30	Pyrazine	2,3-dimethylpyrazine *	7.85	1352	Musty, nut skins, cocoa powdery and roasted with potato and coffee nuances
31	Alcohol	1-hexanol	7.89	1355	Pungent, ethereal, fusel oil, fruity and alcoholic, sweet with a green top note
32	Alcohol	3-octanol	8.42	1396	Earthy, mushroom, dairy, musty, creamy, waxy with a slight fermented green minty nuance
33	Alkane	tetradecane	8.47	1400	Mild waxy
34	Aldehyde	nonanal	8.48	1401	Waxy, aldehydic, citrus, with a fresh slightly green lemon peel like nuance, and a cucumber fattiness
35	Pyrazine	2-ethyl-methylpyrazine *	8.59	1408	Coffee bean, nutty, grassy, roasted
36	Ketone	3-octen-2-one	8.71	1416	Earthy, oily, ketonic, sweet, with hay and mushroom nuances
37	Alkane	3-ethyl-2-methyl-1,3-hexadiene	8.91	1429	
38	Aldehyde	2-octenal	9.04	1437	Fatty, green, herbal
39	Pyrazine	2,6-diethylpyrazine *	9.12	1443	Nutty, hazelnut
40	Alcohol	1-octen-3-ol	9.27	1453	Earthy, green, oily, vegetative and fungal
41	Pyrazine	3-ethyl-2,5-dimethylpyrazine *	9.30	1455	Potato, cocoa, roasted, nutty
42	Sulfur compound	methional *	9.35	1458	Creamy tomato, potato skin and French fry, yeasty, bready, limburger cheese with a savory meaty brothy nuance
43	Pyrazine	2-ethyl-3,5-dimethylpyrazine *	9.52	1469	Peanut, nutty, caramel, coffee, musty, cocoa, pyrazine and roasted
44	Pyrazine	tetramethylpyrazine *	9.53	1470	Nutty, musty and vanilla with dry, brown cocoa nuances
45	Pyrazine	2-methyl-5-propylpyrazine *	9.71	1482	
46	Alcohol	2-ethyl-1-hexanol	9.89	1494	Citrus, fresh, floral, oily, sweet
47	Alkane	pentadecane	9.99	1500	Waxy
48	Pyrazine	3,5-diethyl-2-methylpyrazine *	10.07	1505	Nutty, meaty, vegetable
49	Pyrazine	2,3,5-Trimethyl-6-ethylpyrazine *	10.26	1516	
50	Aldehyde	benzaldehyde	10.56	1533	Almond, fruity, powdery, nutty and benzaldehyde-like
51	Pyrazine	2,5-dimethyl-3-isobutylpyrazine *	10.60	1535	
52	Pyrazine	2-acetyl-5-methylfuran *	10.92	1554	Sweet, musty, nutty with a caramellic nuance
53	Alcohol	n-octanol	11.06	1562	Waxy, green, citrus, aldehydic and floral with a sweet, fatty, coconut nuance
54	Alkane	hexadecane	11.72	1600	
55	Ketone	2-undecanone	11.81	1605	Waxy, fruity, ketonic with fatty pineapple nuances
56	Alcohol	2-octen-1-ol	12.11	1621	Green, vegetable
57	Pyrazine	2-isoamyl-6-methylpyrazine *	12.30	1631	
58	Aldehyde	phenylacetaldehyde	12.59	1646	Honey, floral rose, sweet, fermented, chocolate with a slight earthy nuance
59	Pyrazine	pyrazine,2-butyl-3,5-dimethyl *	12.78	1657	Sweet, earthy
60	Ketone	acetophenone	12.82	1658	Sweet, cherry pit, marzipan and coumarinic, vanilla nuance
61	Terpene	pristane	12.98	1667	
62	Pyrazine	2,5-dimethyl-3-cis-propenylpyrazine *	13.16	1677	
63	Alkane	heptadecane	13.60	1700	
64	Aldehyde	2,4-nonadienal	13.74	1707	Fatty, melon, waxy, green, leaf, cucumber, tropical fruit
65	Aldehyde	undecenal	14.68	1756	Fresh, fruity, orange peel
66	Aldehyde	3-dodecen-1-al	14.68	1756	Bitter orange, mandarin, coriander
67	Aldehyde	2,4-decadienal	14.92	1768	Acrid, greasy
68	Terpene	phytane	15.03	1774	
69	Alkane	octadecane	15.53	1800	
70	FAME	methyl laurate	15.65	1806	Waxy, soapy, creamy, coconut, with mushroom nuances
71	Aldehyde	2,4 decadienal	15.81	1814	Fatty, citrus, nutty
72	Alkane	n-nonadecane	17.49	1900	Bland
73	Alcohol	phenylethyl alcohol	17.79	1916	Sweet, floral, fresh and bready with a rosey honey nuance
74	Aldehyde	2-phenyl-2-butenal	18.23	1939	Musty, floral, honey, powdery and cocoa
75	Heterocyclic compound	2-acetylpyrrole *	18.71	1964	Musty, nutty-like with a coumarin nuance
76	Alcohol	1-dodecanol	18.89	1973	Earthy, soapy, waxy, fatty, honey, coconut
77	Alkane	eicosane	19.41	2000	Waxy
78	FAME	methyl myristate	19.64	2012	Fatty, waxy, petal
79	Aldehyde	5-methyl-2-phenyl-2-hexenal	20.64	2065	
80	Alkane	heneicosane	21.30	2100	
81	FAME	methyl pentadecanoate	21.58	2116	
82	Alkane	docosane	23.10	2200	
83	FAME	methyl palmitate	23.46	2221	Oily, waxy, fatty, orris
84	Alkane	tetracosane	26.86	2400	
85	FAME	methyl stearate	27.02	2413	Oily, waxy
86	FAME	methyl oleate	27.36	2440	Mild, fatty
87	FAME	methyl linoleate	28.15	2503	Oily, fatty, woody

Compounds tentatively identified by Wiley 275 and NIST 98 mass spectral libraries; FAME: fatty acid methyl ester; *: identified in roasted samples.

**Table 2 foods-10-01287-t002:** (**a**) Classes of volatiles and toluene (μg/g f.w. equivalents of 4-methyl-2-pentanol) in *P. eryngii* (strains CS1, CS2, CS3 and LGAM106) mushrooms grown on different cultivation substrates (WS, GM and OL). (**b**) Classes of volatiles and toluene (μg/g f.w. equivalents of 4-methyl-2-pentanol) in *P. ostreatus* (strains CS4, CS5, CS6 and LGAM3002) mushrooms grown on different cultivation substrates (WS, GM and OL).

(**a**)												
	***P. eryngii***											
	**CS1**			**CS2**			**CS3**			**LGAM106**		
	**WS**	**GM**	**OL**	**WS**	**GM**	**OL**	**WS**	**GM**	**OL**	**WS**	**GM**	**OL**
Total *	980.6 ± 134.3 a	946.12 ± 171.9 a	1220 ± 463 a	527.8 ± 72.6 a	611.6 ± 278.3 a	446.3 ± 237.1 a	805.0 ± 95.8 a	967.9 ± 169.2 a	1123 ± 274 a	1268 ± 104 a	1177 ± 83 a	1512 ± 173 a
Alkanes	8.98 ± 0.41 a	17.79 ± 7.75 a	17.97 ± 3.12 a	16.69 ± 8.44 a	13.25 ± 5.19 a	20.99 ± 5.65 a	16.82 ± 6.04 a	29.83 ± 18.63 a	26.09 ± 22.91 a	9.84 ± 7.36 a	51.34 ± 3.98b	19.61 ± 6.78 a
Aldehydes	198.1 ± 77.0 a	160.6 ± 42.0 a	115.0 ± 36.9 a	69.21 ± 7.87 a	57.40 ± 14.45 a	45.57 ± 20.95 a	158.3 ± 20.3 a	155.6 ± 27.0 a	98.86 ± 55.55 a	196.2 ± 24.9 a	131.8 ± 3.1 a	111.3 ± 46.2 a
Alcohols	611.9 ± 47.1 a	590.9 ± 127.3 a	896.4 ± 354.1 a	353.1 ± 43.1 a	449.2 ± 225.3 a	325.7 ± 192.5 a	508.8 ± 129.8 a	598.9 ± 156.1 a	790.0 ± 220.5 a	904.6 ± 82.4 a	798.4 ± 21.2 a	1175 ± 114b
FAME	56.07 ± 15.55 a	58.02 ± 29.42 a	79.88 ± 37.08 a	22.09 ± 22.89 a	25.48 ± 20.26 a	13.80 ± 19.52 a	39.49 ± 23.78 a	91.32 ± 15.86 ab	122.7 ± 46.5c	40.25 ± 18.34 a	128.3 ± 49.9 a	118.5 ± 58.7 a
Ketones	86.12 ± 9.38 a	99.08 ± 30.40 a	83.23 ± 35.87 a	48.60 ± 15.47 a	49.93 ± 26.42 a	30.47 ± 14.40 a	64.83 ± 1.44 a	74.15 ± 17.03 a	63.00 ± 14.96 a	95.20 ± 14.68 b	54.47 ± 1.91 a	74.06 ± 17.55 ab
Furans	0.56 ± 0.00	0.53 ± 0.11	n.d.	0.30 ± 0.08	n.d.	n.d.	0.17 ± 0.02	0.33 ± 0.05	n.d.	0.49 ± 0.21	n.d.	n.d.
Toluene	18.25 ± 3.35 a	18.04 ± 12.40 a	25.88 ± 11.05 a	17.29 ± 2.03 a	15.84 ± 9.03 a	9.78 ± 4.61 a	15.45 ± 4.20 a	13.86 ± 5.93 a	15.71 ± 6.96 a	21.37 ± 16.34 a	8.87 ± 0.00 a	20.02 ± 4.12 a
Terpenes	n.d.	0.45 ± 0.21	0.34 ± 0.13	n.d.	n.d.	n.d.	n.d.	13.72 ± 0.27	26.64 ± 6.53	n.d.	3.99 ± 2.39	n.d.
C8 compounds	747.1 ± 68.3 a	734.9 ± 157.0 a	1033 ± 415 a	414.7 ± 55.9 a	509.2 ± 256.2 a	363.5 ± 208.9 a	623.3 ± 142.3 a	715.1 ± 175.5 a	906.6 ± 268.5 a	1065 ± 59 ab	894.9 ± 16.8 a	1311 ± 164 b
(**b**)												
	***P. ostreatus***											
	**CS4**			**CS5**			**CS6**			**LGAM3002**		
	**WS**	**GM**	**OL**	**WS**	**GM**	**OL**	**WS**	**GM**	**OL**	**WS**	**GM**	**OL**
Total *	814.0 ± 45.8 b	1403 ± 52 c	424.3 ± 45.0 a	2232 ± 70 b	2327 ± 23 b	1986 ± 42 a	2118 ± 75 a	2159 ± 40 a	1545 ± 547 a	1436 ± 362 a	2035 ± 217 a	1728 ± 70 a
Alkanes	15.21 ± 0.45 a	11.80 ± 0.73 b	9.91 ± 1.27 b	31.01 ± 0.36 a	26.29 ± 1.77 a	27.83 ± 11.67 a	27.66 ± 8.66 a	11.91 ± 1.47 a	23.74 ± 18.80 a	10.75 ± 0.89 a	27.59 ± 1.44 b	24.00 ± 3.45 b
Aldehydes	68.20 ± 6.27 b	131.9 ± 2.1 a	17.54 ± 0.33 c	104.7 ± 5.8 b	127.5 ± 3.2 c	88.06 ± 1.26 a	73.80 ± 7.47 a	63.89 ± 0.73 a	66.88 ± 11.21 a	84.56 ± 6.57 a	82.16 ± 14.05 a	57.87 ± 3.45 a
Alcohols	504.3 ± 29.8 b	1018 ± 84 a	263.9 ± 6.7 c	1485 ± 20 b	1689 ± 3 c	1431 ± 3 a	1647 ± 11 a	1659 ± 19 a	1058 ± 369 a	1043 ± 303 a	1475 ± 177 a	1129 ± 21 a
FAME	30.47 ± 2.70 a	6.92 ± 0.41 b	5.19 ± 0.68 b	20.50 ± 2.50 a	7.41 ± 1.54 b	6.27 ± 0.08 b	9.91 ± 0.35 a	18.62 ± 0.44 b	7.35 ± 1.74 a	17.44 ± 9.62 a	10.85 ± 1.29 a	26.05 ± 2.79 a
Ketones	130.4 ± 9.7 a	168.5 ± 10.0 a	68.19 ± 18.20 b	391.6 ± 34.2 a	351.8 ± 24.9 a	331.1 ± 30.1 a	244.7 ± 26.2 a	294.0 ± 29.1 a	298.1 ± 122.7 a	228.9 ± 0.2 a	309.1 ± 47.3 ab	385.9 ± 62 c
Furans	n.d.	0.94 ± 0.44	n.d.	0.84 ± 0.64	n.d.	0.62 ± 0.15	n.d.	n.d.	0.64 ± 0.45	0.47 ± 0.49	0.56 ± 0.14	n.d.
Toluene	65.30 ± 16.30 a	65.30 ± 19.37 a	59.51 ± 21.04 a	194.0 ± 10.4 b	123.5 ± 9.2 a	98.86 ± 19.51 a	114.1 ± 21.6 a	110.4 ± 7.3 a	90.37 ± 24.33 a	99.15 ± 0.13 a	126.1 ± 23.3 a	104.6 ± 19.3 a
Terpenes	n.d.	n.d.	n.d.	3.75 ± 1.77 a	2.22 ± 0.70 a	1.79 ± 0.39 a	0.09 ± 0.04	n.d.	0.19 ± 0.00	2.33 ± 0.02 ab	4.01 ± 1.41 b	0.44 ± 0.12 a
C8 compounds	654.6 ± 21.1 b	1253 ± 74 c	323.0 ± 20.9 a	1926 ± 54 b	2087 ± 3 c	1796 ± 27 a	1912 ± 48 a	1988 ± 51 a	1376 ± 507 a	1309 ± 322 a	1818 ± 229 a	1529 ± 50 a

*: corresponds to the sum of identified compounds listed in Table 1; Lack of letters in common indicates statistically significant differences (Duncan’s *t*-test. *p* < 0.05 in comparisons of treatment means between different substrates. Results are averages ± SD of three replicates. Abbreviations f.w.; fresh weight, n.d.; not detected, CS: commercial strains; LGAM: Laboratory of General and Agricultural Microbiology (Agricultural University of Athens, Greece); FAME: fatty acid methyl esters; WS, wheat straw; GM, wheat straw with grape marc (1:1 *w*/*w*); OL, olive leaves with olive mill wastes (3:1 *w*/*w*).

**Table 3 foods-10-01287-t003:** Volatile compounds (μg/g f.w. equivalents of 4-methyl-2-pentanol) in raw and roasted mushrooms of *P. eryngii* strain CS3 and *P. ostreatus* strain LGAM3002 cultivated on wheat straw.

Classes of Compounds		Rt	*P. eryngii*		*P. ostreatus*	
			Raw	Roasted	Raw	Roasted
	Water content (%)		90.57 ± 1.91	46.12 ± 2.8	90.95 ± 2.1	44.9 ± 3.1
	Total lipids (mg/g f.w.)		7.29 ± 0.4	--	3.73 ± 0.2	--
	Total volatiles (μg/g f.w.)		924.2 ± 233.3	398.1 ± 82.2	1165 ± 129	364.7 ± 22.0
Alkanes	undecane	5.11	25.88 ± 3.79	17.07 ± 5.76	16.98 ± 7.10	12.39 ± 5.02
	dodecane	6.11	1.63 ± 0.12	n.d.	3.66 ± 3.65	n.d.
	pentadecane	9.99	8.47 ± 0.46	n.d.	5.74 ± 4.90	n.d.
	hexadecane	11.72	8.03 ± 1.06	n.d.	4.72 ± 3.14	n.d.
	heptadecane	13.60	6.26 ± 0.78	n.d.	3.83 ± 2.37	n.d.
	octadecane	15.53	5.61 ± 0.27	n.d.	3.15 ± 1.82	n.d.
	nonadecane	17.49	5.42 ± 0.94	n.d.	1.92 ± 0.95	n.d.
	eicosane	19.41	4.27 ± 0.39	n.d.	1.28 ± 0.36	n.d.
Alcohols	1-hexanol	7.89	2.37 ± 0.24	n.d.	6.09 ± 5.51	n.d.
	3-octanol	8.42	9.39 ± 7.27	6.32 ± 3.04	330.6 ± 75.9	65.42 ± 26.73
	1-octen-3-ol	9.27	491.6 ± 208.5	178.3 ± 29.1 *	341.4 ± 131.5	90.19 ± 16.59
	2-ethyl-1-hexanol	9.89	1.68 ± 0.34	0.99 ± 0.44	3.06 ± 1.44	n.d.
	1-octanol	11.06	23.26 ± 14.56	2.98 ± 1.22	9.47 ± 6.45	0.59 ± 0.03
	phenylethyl alcohol	17.79	1.83 ± 0.38	n.d.	n.d.	n.d.
Aldehydes	2-methylbutanal	3.17	n.d.	7.70 ± 9.16	n.d.	n.d.
	3-methylbutanal	3.29	n.d.	14.94 ± 2.35	n.d.	7.10 ± 1.55
	pentanal	3.89	n.d.	n.d.	n.d.	15.73 ± 6.19
	hexanal	4.98	78.91 ± 25.55	42.08 ± 27.17	17.26 ± 7.81	5.50 ± 2.03
	trans-2-hexenal	6.37	1.59 ± 0.39	n.d.	0.39 ± 0.19	n.d.
	octanal	7.14	8.04 ± 2.42	n.d.	4.89 ± 2.53	1.68 ± 1.10
	2-heptenal	7.60	28.93 ± 15.73	n.d.	10.32 ± 1.61	n.d.
	2-octenal	9.04	42.12 ± 16.00	15.47 ± 17.66	14.89 ± 4.34	1.17 ± 0.14
	benzaldehyde	10.56	15.40 ± 6.16	8.71 ± 4.57	3.60 ± 3.21	4.21 ± 1.62
	phenylacetaldehyde	12.59	n.d.	5.61 ± 2.88	n.d.	12.91 ± 6.05
	2,4-nonadienal	13.74	11.13 ± 7.91	n.d.	1.53 ± 0.68	n.d.
	2,4-decadienal (*E,E*)	15.81	2.92 ± 1.43	n.d.	1.73 ± 0.26	n.d.
	2-phenyl-2-butenal	17.92	n.d.	n.d.	n.d.	1.87 ± 1.15
	5-methyl-2-phenyl-2-hexenal	20.64	n.d.	n.d.	n.d.	1.39 ± 0.79
FAME	methyl laurate	15.65	n.d.	n.d.	1.96 ± 0.11	n.d.
	methyl myristate	19.64	3.21 ± 1.86	n.d.	2.50 ± 1.10	n.d.
	methyl pentadecanoate	21.58	4.67 ± 3.45	n.d.	7.83 ± 5.64	n.d.
	methyl palmitate	23.46	34.14 ± 33.47	2.06 ± 0.48	13.29 ± 5.59	0.77 ± 0.27
	methyl stearate	27.02	1.49 ± 0.16	n.d.	n.d.	n.d.
	methyl oleate	27.36	21.11 ± 20.21	1.32 ± 0.18	2.31 ± 1.09	n.d.
	methyl linoleate	28.15	18.20 ± 15.98	1.52 ± 0.22	10.67 ± 4.05	0.64 ± 0.25
Ketones	2-heptanone	5.95	n.d.	n.d.	n.d.	15.28 ± 13.38
	3-octanone	6.75	36.80 ± 5.36	9.53 ± 5.71	310.7 ± 125.3	25.08 ± 12.30
	2-octanone	7.09	4.68 ± 2.52	6.60 ± 2.21	n.d.	7.31 ± 9.03
	2,3-octanedione	7.50	n.d.	n.d.	1.21 ± 0.37	n.d.
	3-octen-2-one	8.71	7.04 ± 6.87	n.d.	n.d.	n.d.
Aromatic compounds	toluene	4.58	14.22 ± 5.26	60.73 ± 11.50	90.62 ± 12.05	27.20 ± 8.27
Pyrazines	methylpyrazine	6.83	n.d.	n.d.	n.d.	0.63 ± 0.18
	2,5-dimethylpyrazine	7.60	n.d.	4.33 ± 0.86	n.d.	5.44 ± 3.39
	2,6-dimethylpyrazine	7.69	n.d.	1.92 ± 1.14	n.d.	3.09 ± 1.94
	ethylpyrazine	7.75	n.d.	n.d.	n.d.	0.79 ± 0.51
	2,3-dimethylpyrazine	7.85	n.d.	0.89 ± 0.41	n.d.	0.78 ± 0.56
	2-ethyl-methylpyrazine	8.59	n.d.	3.69 ± 0.43	n.d.	4.45 ± 2.98
	2,6-diethylpyrazine	9.12	n.d.	n.d.	n.d.	1.76 ± 0.88
	3-ethyl-2,5-dimethylpyrazine	9.30	n.d.	8.76 ± 0.31	n.d.	21.22 ± 5.11
	2-ethyl-3,5-dimethylpyrazine	9.52	n.d.	2.85 ± 0.10	n.d.	6.00 ± 2.36
	tetramethylpyrazine	9.53	n.d.	n.d.	n.d.	0.56 ± 0.06
	2-methyl-5-propylpyrazine	9.71	n.d.	n.d.	n.d.	0.67 ± 0.25
	3,5-diethyl-2-methyl pyrazine	10.07	n.d.	n.d.	n.d.	5.94 ± 1.85
	2,3,5-trimethyl-6-ethylpyrazine	10.40	n.d.	n.d.	n.d.	2.40 ± 0.47
	2,5-dimethyl-3-isobutylpyrazine	10.60	n.d.	n.d.	n.d.	1.81 ± 0.51
	2-isoamyl-6-methylpyrazine	12.30	n.d.	n.d.	n.d.	2.00 ± 0.49
	2-butyl-3,5-dimethyl pyrazine	12.94	n.d.	n.d.	n.d.	4.49 ± 1.23
	2,5-dimethyl-3-propenylpyrazine	13.27	n.d.	n.d.	n.d.	1.15 ± 0.65
Sulfur compounds	dimethyl disulfide	4.82	n.d.	n.d.	n.d.	2.67 ± 2.19
	methional	9.35	n.d.	1.96 ± 0.36	n.d.	2.59 ± 1.62
Terpenes	limonene	6.16	5.21 ± 0.56	1.07 ± 0.41	0.30 ± 0.01	n.d.
	pristane	12.98	0.18 ± 0.02	n.d.	4.32 ± 3.47	n.d.
	phytane	15.03	n.d.	n.d.	3.94 ± 1.49	n.d.
Other heterocyclic compounds	2,4,5-trimethyloxazole	6.07	n.d.	n.d.	n.d.	1.72 ± 1.48
	2-pentyl-furan	6.47	4.43 ± 2.31	1.50± 0.32	1.05 ± 0.48	0.51 ± 0.36
	2-acetyl-5-methylfuran	10.92	n.d.	3.00 ± 0.45	n.d.	n.d.
	2-acetylpyrrole	18.71	n.d.	n.d.	n.d.	1.66 ± 0.56

* Results are presented as averages ± SD of three replicates; n.d.; not detected, CS: commercial strains; LGAM: Laboratory of General and Agricultural Microbiology (Agricultural University of Athens, Greece); FAME: fatty acid methyl esters.

## Data Availability

There are no Data Availability Statements from the present study.

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
