# Peer review of "Volatile Profiling of Pleurotus eryngii and Pleurotus ostreatus Mushrooms Cultivated on Agricultural and Agro-Industrial By-Products"

_foods, 2021, doi:10.3390/foods10061287_

Round 1
Reviewer 1 Report
Dear Authors,
The submited manuscript is well-writen and very interesting. However some issue should be cliarified:
1) Why in some results SD is so high?
2) In my opinion for some results mg/g should be used as a unit, because some presented results are difficult to read (line 163, 196 etc; Table 3 total volatiles)
Author Response
Reviewer #1
Query No 1: Why in some results SD is so high?
Response: Analyses were conducted in fresh mushrooms that were sampled and frozen immediately after harvest, as it is described in paragraph 2.3. Given that even for mushrooms from the same batch, differences in the samples analysed might occur, since they could have derived (for example) from different part of mushrooms (i.e., stipe or pileus). Therefore, every measurement is unique thus affecting the final result and leading to increased variance.
Query No 2: In my opinion for some results mg/g should be used as a unit, because some presented results are difficult to read (line 163, 196 etc; Table 3 total volatiles)
Response: We agree with the reviewer regarding the difficulty to read the results in some cases. To make the tables more reader-friendly we reduced the number of decimal digits throughout manuscript.
Reviewer 2 Report
line 157 – the truth of this statement is based on the threshold of detection limits for your study, therefore the statement should be modified to indicated that the number of volatiles above the threshold of detection in this system…
There are other issue with this statement that also need to be considered. It assumes that those volatiles are at a concentration above the human threshold of detection. Are they? You assume that all of those volatiles are desirable volatiles. Are they? You assume that the concentrations of the most important volatiles are higher just because more total volatiles are present – are they? This discussion needs to be revised to take these considerations into account.
Lines 162 sounds like it contradicts 156. Revise 162 to make this clear, ie Nevertheless, concentrations of the volatiles (i.e., compounds included in Tables 1 and 3) were higher in P. ostreatus compared to P. eryngii
The discussion tends to assume that the number of volatiles is the most important topic to discuss. In terms of flavor, which is what is actually important, what matters is how much the concentration is above the human threshold of detection. The discussion needs to acknowledge that the concentration above the limit of detection is what is important, and focus on this in the discussion.
Discussion by volatile type is hard to follow. It sounds like a dissertation, explaining the derivation of each volatile. If the discussion is going to be grouped by volatile type, it needs to focus on explaining how the mushroom species or substrate affects the formation of these volatile groups. Why are these different by group and why is this important?
The effect of substrate appears to be the focus, based on the title, but there is little discussion of this. Line 441 concludes with a statement about using the work to understand organleptic properties. There was not a lot of discussion to back that statement. More importantly, the conclusion did not discuss the implications of the effect of substrate, which is I assume was that these substrates could be used to produce mushrooms with no loss of flavor
Author Response
Reviewer #2
Query No 1: line 157 – the truth of this statement is based on the threshold of detection limits for your study, therefore the statement should be modified to indicate that the number of volatiles above the threshold of detection in this system…
Response: Among the volatile compounds detected only those fulfilling the criteria set at lines 127-129 of Materials & Methods were finally included in the respective tables and discussed. This is stressed now with a short comment added in line 157.
Query No 2: There are other issue with this statement that also need to be considered. It assumes that those volatiles are at a concentration above the human threshold of detection. Are they?
Response: The odor threshold values of volatile compounds were not taken into account for including them in the respective tables. As it is stated in lines 127-129 of Materials & Methods, only those compounds exhibiting mass spectra matching qualities higher than 90% and calculated RIs not differing by more than ±15 from values available in public domain databases were considered and discussed.
Query No 3: You assume that all of those volatiles are desirable volatiles. Are they?
Response: No, we do not think that all those volatiles are desirable, we just refer to the volatiles that were totally detected
Query No 4: You assume that the concentrations of the most important volatiles are higher just because more total volatiles are present– are they?
Response: We cannot make such an assumption.
Query No 5: This discussion needs to be revised to take these considerations into account.
Response: Since the sentence in lines 157-159 can lead to false impressions, it was removed, together with Reference no 39 and references were renumbered accordingly.
Query No 6: Lines 162 sounds like it contradicts 156. Revise 162 to make this clear, ie Nevertheless, concentrations of the volatiles (i.e., compounds included in Tables 1 and 3) were higher in P. ostreatus compared to P. eryngii
Response: Actually there is no contradiction, as text in line 156 refers to the number of compounds detected in the present study, while text in line 162 (now 167) refers to the sum of their concentrations, thus to the total concentration of the aroma volatile compounds. To make the expression more clear text in line 162 was modified according to the reviewer’s suggestion.
Query No 7: The discussion tends to assume that the number of volatiles is the most important topic to discuss. In terms of flavor, which is what is actually important, what matters is how much the concentration is above the human threshold of detection. The discussion needs to acknowledge that the concentration above the limit of detection is what is important, and focus on this in the discussion.
Response: In some parts on the text we do mention the human thresholds of detection –for example in lines 181, 210, 230 and in paragraph 3.1– indicating in this way the difference in the perception of the aroma compounds. However, the main purpose of this study was to test if the volatile profiles can differentiate among the mushrooms species or strains and/or substrates used, and now this is more clearly stated in Introduction (lines 70-72). In addition a comment regarding the significance of odor thresholds was added in lines 163-166.
Query No 8: Discussion by volatile type is hard to follow. It sounds like a dissertation, explaining the derivation of each volatile. If the discussion is going to be grouped by volatile type, it needs to focus on explaining how the mushroom species or substrate affects the formation of these volatile groups. Why are these different by group and why is this important?
Response: In the existing literature presenting the results grouped as classes of volatiles is the trend and we followed it. To our opinion, if the results were presented compound by compound, it would be difficult for the reader to follow the discussion.
Regarding the explanation of how the mushroom species or substrate affect the formation of these volatile classes, and why these are different by group, and why is this important, we did not focus on these topics because they are outside the aim of this study as aforementioned.
Query No 9: The effect of substrate appears to be the focus, based on the title, but there is little discussion of this. Line 441 concludes with a statement about using the work to understand organoleptic properties. There was not a lot of discussion to back that statement. More importantly, the conclusion did not discuss the implications of the effect of substrate, which is I assume was that these substrates could be used to produce mushrooms with no loss of flavor
Response: We agree with the reviewer, and we removed the last sentence of Conclusions (lines 450-451).
The mushrooms used in the present study were cultivated on agro-industrial by-products and not exclusively on the commonly used substrate, namely wheat straw. The main objective of this study was the volatiles’ profiling of mushrooms produced by different species/strains on various substrates in order to examine whether (and to what extent) these were affected and if discrimination could be possible among them.
The results demonstrated that volatiles are separated/grouped per species but not on the basis of the substrates used. The latter finding is interesting considering that previous studies on the same mushrooms revealed an influence of substrates on the antioxidant activity, β-glucans content and the levels of bioactive microconstituents like ergosterol, phenolic and terpenic acids and free amino acids (Koutrotsios et al 2018; Tagkouli et al. 2020). This comment was added in Conclusions (lines 447-450).
References:
- Koutrotsios, G.; Kalogeropoulos, N.; Kaliora, A.C.; Zervakis, G.I. Toward an increased functionality in oyster (Pleurotus) mushrooms produced on grape marc or olive mill wastes serving as sources of bioactive compounds. Agric. Food Chem. 2018, 66, 5971–5983.
- Tagkouli, D.; Kaliora, A.; Bekiaris, G.; Koutrotsios, G.; Christea, M.; Zervakis, G.I.; Kalogeropoulos, N. Free amino acids in three Pleurotus species cultivated on agricultural and agro-industrial by-products. Molecules 2020, 25, 4015.